# Control and Realism: Best of Both Worlds in Layout-to-Image without Training

Bonan Li [* 1 2]   Yinhan Hu [* 1]   Songhua Liu [2]   Xinchao Wang [2]

## Abstract

Layout-to-Image generation aims to create complex scenes with precise control over the placement and arrangement of subjects. Existing works have demonstrated that pre-trained Text-to-Image diffusion models can achieve this goal without training on any specific data; however, they often face challenges with imprecise localization and unrealistic artifacts. Focusing on these drawbacks, we propose a novel training-free method, ***WinWinLay***. At its core, WinWinLay presents two key strategies—Non-local Attention Energy Function and Adaptive Update—that collaboratively enhance control precision and realism. On one hand, we theoretically demonstrate that the commonly used attention energy function introduces inherent spatial distribution biases, hindering objects from being uniformly aligned with layout instructions. To overcome this issue, non-local attention prior is explored to redistribute attention scores, facilitating objects to better conform to the specified spatial conditions. On the other hand, we identify that the vanilla backpropagation update rule can cause deviations from the pre-trained domain, leading to out-of-distribution artifacts. We accordingly introduce a Langevin dynamics-based adaptive update scheme as a remedy that promotes in-domain updating while respecting layout constraints. Extensive experiments demonstrate that WinWinLay excels in controlling element placement and achieving photorealistic visual fidelity, outperforming the current state-of-the-art methods.

---

[*]Equal contribution   [1]University of Chinese Academy of Sciences, Beijing, China [2]National University of Singapore, Singapore. Correspondence to: Xinchao Wang <xinchao@nus.edu.sg>.

*Proceedings of the $42^{nd}$ International Conference on Machine Learning*, Vancouver, Canada. PMLR 267, 2025. Copyright 2025 by the author(s).

## 1. Introduction

Recent advances in Text-to-Image (T2I) generation (Rombach et al., 2022; Podell et al., 2023) have profoundly revolutionized the vision landscape, facilitating the synthesis of highly authentic assets from textual prompts, *e.g.*, text-driven Image-to-Image translation (Tumanyan et al., 2023; Parmar et al., 2023; Ruiz et al., 2023; Shi et al., 2024; Tosi et al., 2025) and video generation (Wu et al., 2023; Zhang et al., 2023; Jiang et al., 2024; Qing et al., 2024; Kwon et al., 2025). Nevertheless, designing comprehensive prompts to meticulously control every aspect of an image can be both labor-intensive and time-consuming, posing challenges for efficient generation workflows. As a remedy, Layout-to-Image (L2I) models (Xue et al., 2023; Zheng et al., 2023; Chen et al., 2024b; Liu et al., 2024a) have been developed, guiding the process in a desired direction by incorporating user-provided inputs, such as bounding boxes.

To acquire L2I models, an intuitive framework is to fine-tune powerful T2I models with with spatial conditioning. However, these approaches (Li et al., 2023; Wu et al., 2024) incurs expensive training cost and faces challenges when collecting resource-intensive paired data. To overcome the aforementioned limitations, existing methods (Couairon et al., 2023; Xie et al., 2023) have explored the incorporation of layout guidance during the sampling phase, establishing the training-free paradigm for L2I. Among them, backward guidance (Chen et al., 2024d), which combines attention redistribution and backpropagation update rules, has been demonstrated as a promising scheme (Liu et al., 2024a; Chen et al., 2024b). Specifically, attention redistribution encourages cross-attention activation at designated positions via the energy function, while the backpropagation rule directly updates feature maps using corresponding gradients to match the specified layout. However, while these works offer higher efficiency, they fall short of training-based approaches in terms of spatial fidelity and image quality.

To bridge this gap, we initiate our exploration with an in-depth analysis of existing mechanisms and demonstrate that, although effective, they are still prone to suboptimal results in terms of control capabilities and visual realism. Firstly, we theoretically verify that the ***attention energy function tends to favor patches with higher initial attention*** within the bounding box during the optimization process, lead-

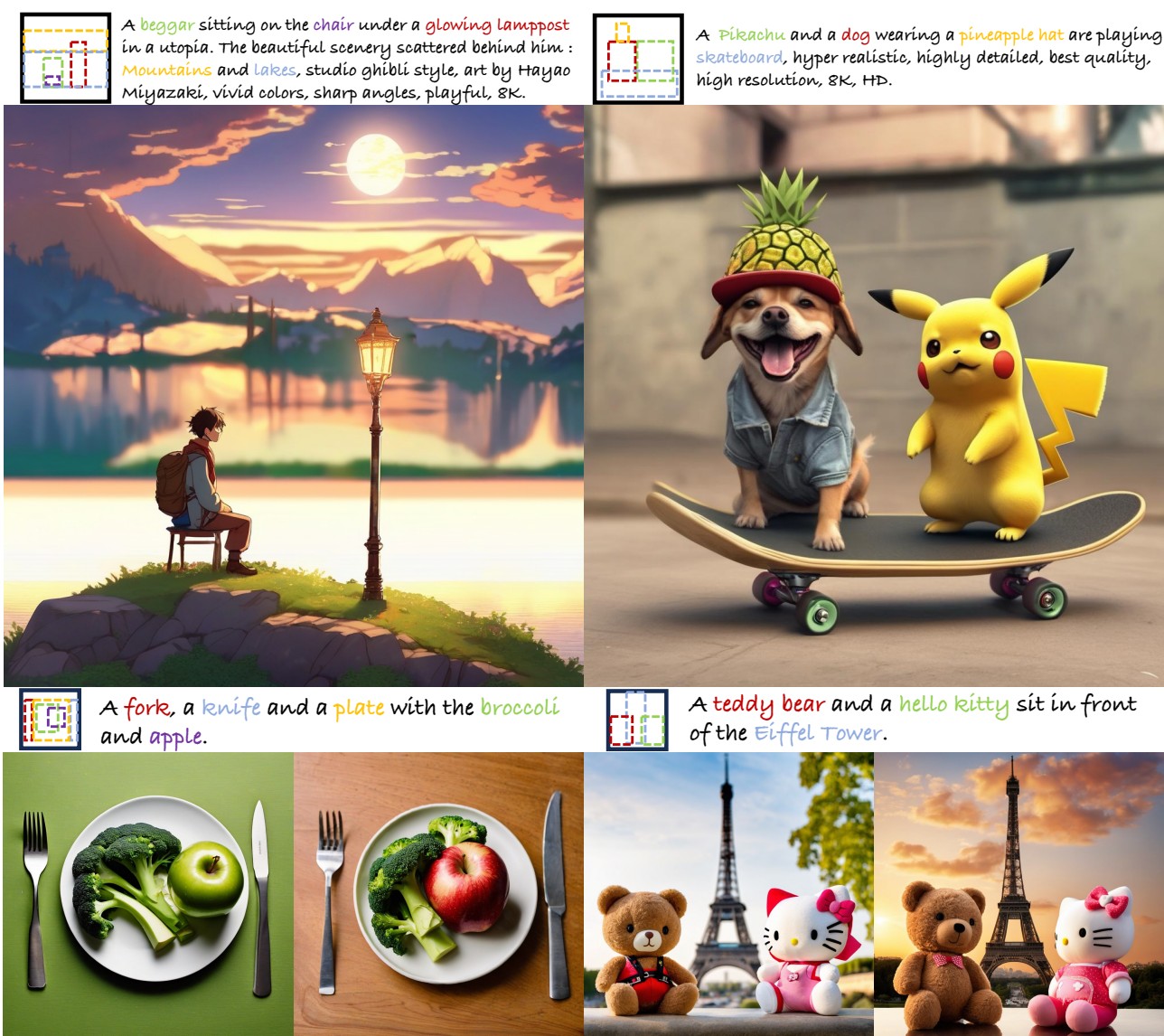

A beggar sitting on the chair under a glowing lamppost in a utopia. The beautiful scenery scattered behind him : Mountains and lakes, studio ghibli style, art by Hayao Miyazaki, vivid colors, sharp angles, playful, 8K.

A Pikachu and a dog wearing a pineapple hat are playing skateboard, hyper realistic, highly detailed, best quality, high resolution, 8K, HD.

A fork, a knife and a plate with the broccoli and apple.

A teddy bear and a hello kitty sit in front of the Eiffel Tower.

Figure 1: Given user-provided bounding boxes and prompts of subjects, our **WinWinLay** generates controllable and realistic images with pre-trained diffusion model, such as SDXL (Podell et al., 2023), without any finetuning with paired data.

ing to the final generated objects being confined to smaller regions rather than evenly distributed across the specified box. Secondly, the ***backpropagation update rule fails to account for the pre-trained distribution when biasing features***, resulting in a trade-off between control and image quality, where stronger control often comes at the cost of poorer visual appearance. To the best of our knowledge, this is the first study that theoretically analyze these two core components of backward guidance in Layout-to-Image task.

In this paper, we propose a novel model named WinWinLay, which generates high-quality training-free L2I results by explicitly accounting for above challenges. On one hand, to better align the given spatial guidance, we augment the attention energy function with the non-local attention prior to encourage attention to uniformly cover the specified region. Additionally, to avoid constraining irregularly shaped objects into a rigid box-like form (*e.g.*, coconut trees usually have broad leaves and slender trunks), we introduce a decaying schedule that gradually decreases the strength of prior along the denoising step that facilitates natural structure. On the other hand, focusing on the trade-off between controllability and realism, we design the update rule based on Langevin dynamics, which brings the best of two worlds by simultaneously incorporating the updating directions given by both layout controls and the original T2I model. Specifically, we introduce an adaptive weighting strategy to balance the two directions across different sampling steps, eliminating the need for cumbersome hyperparameter search.. Experiments demonstrate that the proposed approach mitigates

the above issues successfully and generates satisfactory L2I results (see Figure 1) in a training-free manner.

In summary, our contributions can be summarized as: (*i*) We provide the first theoretical analysis of previous backward guidance methods to the best of our knowledge. Inspired by the theoretical insights, we present an advanced approach, WinWinLay, for Layout-to-Image generation which exhibits significantly control and realistic quality. (*ii*) We propose a novel Non-local Attention Energy Function that guides the model to better adhere to spatial constraints while preserving the natural structure of objects. (*iii*) We explore a Langevin dynamics-based Adaptive Update scheme to eliminate the trade-off between layout instruction and realistic appearance while maintaining efficiency. (*iv*) We conduct extensive experiments to highlight the effectiveness of WinWinLay for both controllability and quality, thereby advancing the practical application of L2I generation.

## 2. Related Work

### 2.1. Text-to-Image Generation

Diffusion models (Ho et al., 2020) have recently disrupted the longstanding dominance of generative adversarial networks (GANs) (Goodfellow et al., 2014) in image synthesis (Dhariwal & Nichol, 2021; Song et al., 2021a; Ho et al., 2022), further accelerating advancements in Text-to-Image (T2I) generation (Saharia et al., 2022; Rombach et al., 2022; Podell et al., 2023; Peebles & Xie, 2023; Chen et al., 2024c). Benefitting from training on large-scale image-text datasets (Schuhmann et al., 2022b), they exhibited remarkable ability to generate diverse, creative images controlled by text prompts. Moreover, recent developments also unlock the potential of T2I to tackle challenging vision tasks, such as image editing (Brooks et al., 2023; Kawar et al., 2023; Ruiz et al., 2023; Mokady et al., 2023; Xu et al., 2024), style transfer (Sohn et al., 2024; Chen et al., 2024a; Ahn et al., 2024) and 3D generation (Chen et al., 2024e; Li et al., 2024; Wang et al., 2024b). Despite substantial progress, existing works still struggle to precisely control image details, such as layout, which significantly hampers their applicability in real-world scenarios. In this paper, we focus on controlling subject synthesis within the complex scene through user-specified layout condition in a training-free manner.

### 2.2. Layout-to-Image Generation

Layout-to-Image (L2I) (Xue et al., 2023; Zheng et al., 2023; Xie et al., 2023; Jia et al., 2024; Liu et al., 2024b;a) focus on generating images that simultaneously adhere to the textual prompt and corresponding layout instructions, *e.g.*, bounding boxes and scribble. To achieve this, existing works (Li et al., 2023; Yang et al., 2023; Wang et al., 2024a; Zhou et al., 2024) propose to finetune the powerful Text-to-

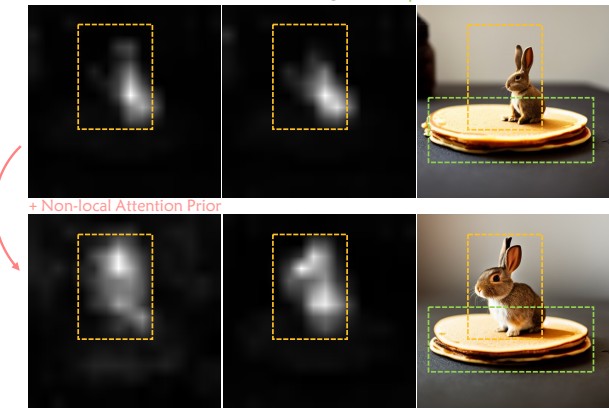

Figure 2: Visualization of cross-attention between text token "rabbit" and intermediate features of the denoiser. It can be observed that the attention energy function leads to attention focusing on a local region. Conversely, after incorporating non-local attention prior, the attention attempts to encompass the entire bounding box. To demonstrate the robustness of this prior, "pancake" is consistently equipped with non-local attention prior in this context.

Image models with paired data. However, collecting such extensive labeled images is not non-trivial and high-cost overheads also limit the usage of L2I in practice. To overcome the aforementioned challenges, recent studies (Bar-Tal et al., 2023; Kim et al., 2023; Singh et al., 2023; Couairon et al., 2023; Chen et al., 2024d;b; Liu et al., 2024a) explore training-free approaches that utilize forward or backward guidance mechanisms within the denoising process. Despite this success, the generated results often deviate from the predefined positions and exhibit severe unrealistic artifacts. Here, we theoretically analyze the backward guidance and propose improved strategies to alleviate these problems.

## 3. Preliminaries

### 3.1. Latent Diffusion Model

Our work recruits the Latent Diffusion Model (LDM) (Rombach et al., 2022) as prior, which defines a generative process that gradually transforms a noise latent $\epsilon$ and a text prompt $p$ to an image $x$. Specifically, LDM first encodes the image $x_0$ into in a low-dimensional latent space using a pre-trained encoder $E$, *i.e.*, $z_0 = E(x_0)$ and operates the diffusion process. Then, the representation $z_0$ are decoded back into the image $x_0$ using a pre-trained decoder $D$. Specifically, given a noised latent $z_t$ at step $t$ and text tokens $y = \phi(p)$ where $\phi$ is a frozen text encoder (Radford et al., 2021), the denoiser $\epsilon_\theta$ is optimized to remove the noise $\epsilon$ added to the latent code $z_0$:

$$\min_\theta \mathbb{E}_{z_0,\epsilon,t}||\epsilon - \epsilon_\theta(z_t, t, y)||_2^2. \tag{1}$$

Here, learnable parameters $\theta$ is typically integrated into a U-Net (Ronneberger et al., 2015) architecture, which comprises convolutional layers, self-attention and cross-attention mechanisms.

### 3.2. Backward Guidance for Layout-to-Image

To precisely position the subject at the specified location, backward guidance (Chen et al., 2024d) aims to sample desired images from the distribution $p(z|y, b, i)$, where the bounding box $b_i$ corresponding to the text token $y_i$ of target subject. Specifically, backward guidance begins by designing an optimizable object, *e.g.*, attention energy function $\mathcal{E}_{aef}$, to redistribute attention map $a$ in feature $z$, encouraging the attention values of the $i^{th}$ token to focus within region $b_i$:

$$\mathcal{E}_{aef}(a^{(\gamma)}, m, i) = \left(1 - \frac{\sum_u m_{ui} \cdot a_{ui}^{(\gamma)}}{\sum_u a_{ui}^{(\gamma)}}\right)^2, \quad (2)$$

where $a_{ui}^{(\gamma)}$ quantifies the strength of the association between each location $u$ in cross-attention layer $\gamma$ and token $y_i$, $m_i$ denotes the binary mask which is transformed from $b_i$ with pixels inside the box region marked as 1 and those outside as 0. To bias attention maps, the gradient of the Equation (2) is computed via backpropagation to update the initial latent feature $\bar{z} = z$:

$$z \leftarrow \bar{z} - \eta \nabla_z \sum_\gamma \sum_i \mathcal{E}_{aef}(a^{(\gamma)}, m, i), \quad (3)$$

where $\eta$ is a hyperparameter used to control the strength.

## 4. Method

In this section, we introduce WinWinLay, a training-free Layout-to-Image generation framework. First, we provide a detailed introduction to Non-local Attention Energy Function (Section 4.1), which is used to enhance layout constrain. Subsequently, we shift focus to explore Adaptive Update (Section 4.2) to eliminate the trade-off between control and quality.

### 4.1. Non-local Attention Energy Function

Attention energy function (Chen et al., 2024d) is a widely used loss term for guiding attention redistribution, but it often leads to objects occupying local region of the bounding box, hindering precise control. To address this, non-local attention prior is introduced to encourage attention to be smoothly distributed across the specified location.

***Revisiting of Attention Energy Function.*** Based on the overview of attention energy function (Section 3.2), we can straightforwardly reformulate Equation (2) as following for

intuitive analysis:

$$\mathcal{E}_{aef}(a, m) = (1 - \sum_u \tilde{a}_u \cdot m_u)^2, \quad (4)$$

where $\tilde{a}_u = a_u / \sum_u a_u$ denotes the normalization of $a_u$ which denotes attention value of $u^{th}$ patch in attention map $a$. Notably, for simplicity and clarity of presentation, the notation for subjects and cross-attention layer is omitted. Given $\max m_u = 1$, so we have $\max_{\tilde{a}} \sum_u \tilde{a}_u \cdot m_u = 1$ and then obtain following equation:

$$\mathcal{E}_{aef}(a, m) = (\max_{\tilde{a}} \sum_u \tilde{a}_u \cdot m_u - \sum_u \tilde{a}_u \cdot m_u)^2. \quad (5)$$

Here, it is evident that minimizing $\mathcal{E}_{aef}$ is equivalent to maximizing $\tilde{a} \cdot m$. Specifically, when $\tilde{a} \cdot m$ attains maximum value, the support set of $\tilde{a}$ is guaranteed to be contained within support set of $m$. Building on this insight, we first observe that the optimal solution to the attention energy function is not unique. Actually, when the support set of $\tilde{a}$ is entirely contained within $m$, $\tilde{a} \cdot m$ can be maximized. However, this non-uniqueness may lead to the support of $\tilde{a}$ concentrating in localized regions of $m$, thereby compromising effective control over the spatial layout. Meanwhile, we notice that $\nabla_{\tilde{a}} \mathcal{E}_{aef}(a, m) \parallel m$, causing all patches within the masked region to receive identical gradient magnitudes. This gives patches with larger initial values a significant advantage during the optimization process, thereby amplifying localized effects (see Figure 2). To support this perspective, we start by considering a simple yet universal optimization objective as follows:

$$\max_v f(v) = m \cdot \text{softmax}(v), where\ v \in \mathbb{R}^n,\ m \in \{0, 1\}^n, \quad (6)$$

and we denote $q = \text{softmax}(v)$ for simplicity.

**Theorem 4.1.** *Assume that during the optimization process at a certain step, there exist $m_j = m_k = 1$ and $q_j > q_k$. After a single gradient update with step size $\beta > 0$, the updated values $q'_j$, $q'_k$ satisfy $q'_j > q'_k$ and $\frac{q'_j}{q'_k} > \frac{q_j}{q_k}$.*

*Proof.* First, Jacobian matrix of $q$ with respect to $v$ can be computed as follows:

$$J = diag(q) - qq^T. \quad (7)$$

According to the chain rule, we have the gradient of $v$ by:

$$\nabla_v f(v) = J^T m = \begin{pmatrix} q_1(m_1 - q^T m) \\ \vdots \\ q_n(m_n - q^T m) \end{pmatrix}. \quad (8)$$

Then, $v^j$ and $v^k$ are updated as:

$$v'_j = v_j + \beta(m_j - q^T m)q_j = v_j + \beta' q_j, \quad (9)$$

$$v'_k = v_k + \beta(m_k - q^T m)q_k = v_k + \beta' q_k, \quad (10)$$

Naturally, combining the above equations gives:

$$
\begin{aligned}
\frac{q'_j}{q'_k} &= \exp(v'_j - v'_k) = \exp(v_j - v_k + \beta'(q_j - q_k)) \\
&= \exp(v_j - v_k) \cdot \exp(\beta'(q_j - q_k)) \\
&= \frac{q_j}{q_k} \cdot \exp(\beta'(q_j - q_k)) > \frac{q_j}{q_k}. \qquad \square
\end{aligned}
\quad (11)
$$

Through the analysis of the above problem, it can be concluded that patches within the masked region with larger initial values tend to amplify their relative prominence during the optimization process, thereby suppressing the growth of other regions. This implies that the attention map redistributed by energy function exhibits an implicit bias, favoring regions with larger initial values. Consequently, it becomes challenging to evenly cover the entire box.

***Non-local Attention Prior.*** In this regard, a simple and effective non-local attention prior is introduced to facilitate global attention responses. Different from intuitively conceivable uniform constrain, this prior promotes the placement of objects near the center of the bounding box while encouraging maximal coverage of the entire region. Concretely, given the bounding box $b$ (width as $W$, height as $H$) and its center point $c = (c_x, c_y)$, the normalized distance from point $u = (u_x, u_y)$ within the masked region $S$ to the center can be calculated as $d_u = \frac{(u_x - c_x)^2}{W} + \frac{(u_y - c_y)^2}{H}$. Accordingly, we construct the prior distribution $\tau_u \propto exp(-\lambda d_u)$, where $\lambda \geq 0$ is used to control the variance of the distribution. This design ensures that points farther from the center are assigned smaller probability values. Subsequently, local bias is alleviated by maximizing the KL divergence between the distribution of the attention within $S$ and the prior $\tau$:

$$\mathcal{R}_{nap} = \sum_{u \in S} \hat{a}_u \log \frac{\hat{a}_u}{\tau_u}, \quad (12)$$

where $\hat{a}_u = a_u / \sum_{u \in S} a_u$ and $a$ denotes attention value.

***Total Loss.*** Here, non-local attention energy function is formulated as the summation of $\mathcal{E}_{aef}$ and $\mathcal{R}_{nap}$ across all subject $i$ and layer $\gamma$:

$$
\begin{aligned}
\mathcal{E}_{naef} = \sum_{\gamma} \sum_{i} \Big[ \big(1 - \frac{\sum_u m_{ui} \cdot a_{ui}^{(\gamma)}}{\sum_u a_{ui}^{(\gamma)}}\big)^2 \\
+ \rho \sum_{ui \in S_i} \hat{a}_{ui}^{(\gamma)} \log \frac{\hat{a}_{ui}^{(\gamma)}}{\tau_{ui}} \Big].
\end{aligned}
\quad (13)
$$

To account for the irregular shape of objects in real-world scenarios, we introduce a hyperparameter $\rho$ that decreases linearly with the denoising timesteps, enabling objects to adapt to naturally structure. Similar to existing work (Chen et al., 2024d), we only reallocate cross-attentions with corresponding tokens in the middle and first up layers.

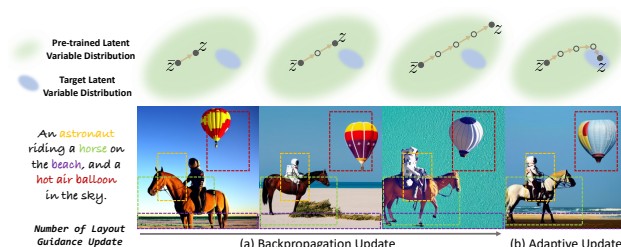

Figure 3: Given the initial feature $\bar{z}$ in a pre-trained distribution, our goal is to iteratively update it within layout constraints to achieve a target latent vector $z$. Here, (a) backpropagation update struggles to effectively maintain spatial control with a limited number of updates, while an increased number of updates often leads to deviations from the pre-trained distribution, leading to a trade-off between control and generation quality. In contrast, (b) adaptive update strategies simultaneously account for the pre-trained distribution and layout constraints, thereby consistently improving overall performance. Zoom in for more details.

## 4.2. Adaptive Update

Although backpropagation update is simple, it fails to balance layout constraints and image quality. Therefore, we propose Adaptive Update to consistently enhance the appearance of outputs based on Langevin dynamics and adaptive distribution construction.

***Revisiting of Backpropagation Update.*** Given latent feature $z_{t+1}$ as timestep $t + 1$, conditional probability $p(z_t | z_{t+1})$ of pretrained diffusion model to obtain an initial estimation $\bar{z}_t$. Subsequently, the gradient update step refines $\bar{z}_t$ to minimize the non-local attention energy function $\mathcal{E}_{naef}$ (replace $\mathcal{E}_{aef}$ for consistent description) to obtained optimized latent variable $z_t = \arg \min \mathcal{E}_{naef}(\bar{z}_t, m)$ via Equation (3) where $a = attention\_map(\bar{z})$. However, update in this manner fails to adequately account for the constraints imposed by the latent variable distribution $p(z_t)$, which may lead to a trade-off between generation control and output quality. Specifically, when the gradient updates are insufficient, the optimization of $\mathcal{E}_{naef}$ remains suboptimal, resulting in poor layout control. Conversely, when the gradient updates are overly exhaustive, the resulting $z_t$ may deviate significantly from the initial estimate $\bar{z}_t$, leading to a reduced likelihood $p(z_t)$. This misalignment adversely affects subsequent denoising steps, ultimately degrading the quality of the generated images. Moreover, the visualization results in Figure 3 under varying levels of control intensity further experimental substantiate this conclusion.

***Langevin Dynamics Updating.*** To eliminate the stubborn trade-off, we propose that both the attention redistribution and $p(z_t)$ should be concurrently considered during the update process. For attention redistribution function, the corresponding Gibbs distribution can be constructed as

$p(m|z_t) \propto exp(-\nu \mathcal{E}_{naef}(z_t, m))$, where $\nu$ is a hyperparameter that controls the shape of the distribution. Given timestep $t$, our ultimate goal is to sample $z_t$ from $p(z_t|m)$. Based on Bayes' Theorem, we have:

$$p(z_t|m) \propto p(z_t)p(m|z_t), \qquad (14)$$

then the score function of $p(z_t|m)$ can be derived as:

$$\nabla_{z_t} \log(p(z_t|m)) = \nabla_{z_t} \log(p(z_t)) + \nabla_{z_t} \log(p(m|z_t))$$
$$= \nabla_{z_t} \log(p(z_t)) - \nu \nabla_{z_t} \mathcal{E}_{naef}(z_t, m). \qquad (15)$$

Here, $\nabla_{z_t} \log(p(z_t))$ represents the unconditional score function which is approximated by pre-trained diffusion model. According to (Song & Ermon, 2019), we can use Langevin dynamics to sample from any distribution with a known score function. Specifically, given a step size $\xi > 0$ and an initial value $\bar{z}_t^{(0)}$, Langevin dynamics iteratively updates as follows:

$$\bar{z}_t^{(k+1)} = \bar{z}_t^{(k)} + \xi \nabla_{\bar{z}_t^{(k)}} \log(p(\bar{z}_t^{(k)}|m)) + \sqrt{2\xi}\epsilon_k, \quad (16)$$

where $\epsilon_k \sim \mathcal{N}(0, I)$ and $0 \leq k \leq O$. As $\xi \to 0$ and $O \to \infty$, the distribution of $\bar{z}_t^O$ will converge to $p(z_t|m)$. Note that for step size $\xi > 0$ and finite $O$, the sampling process can be corrected using the Metropolis-Hastings method to convert it into a strict MCMC sampling procedure. However, this correction step is typically omitted for convenience in practice. Here, similar to (Song et al., 2021b), we determine step size $\xi = 2(r||\epsilon_k||_2 / ||\nabla_{\bar{z}_t} \log(p(\bar{z}_t|m))||_2)^2$ where $r$ is the signal-to-noise ratio.

*Adaptive Distribution Construction.* Although Langevin dynamics effectively mitigates the trade-off, the introduction of the additional hyperparameter $\nu > 0$ of distribution consequently reduces generation efficiency. From Equation (15), it follows that $\nu$ adjusts the weight of $\nabla_{z_t} \mathcal{E}_{naef}(z_t, m)$ in score function $\nabla_{z_t} \log(p(z_t|m))$. Intuitively, a larger $\nu$ results in a steeper distribution, where the optimization process focuses more on minimizing the non-local attention energy function, leading to faster convergence (smaller $O$) but requiring larger step sizes $\xi$ to accelerate the optimization process of $\log(p(z_t))$, which can increase the error of Langevin dynamics and then degrade image quality. Conversely, a smaller $\nu$ produces a flatter distribution, prioritizing image quality preservation, which slows the optimization process (larger $O$) and requires smaller $\xi$ but leads to more iterations, reducing sampling efficiency. Therefore, selecting the appropriate $\nu$ is critical to balance image quality and generation efficiency. Here, we propose treating Equation (16) as a multi-task optimization problem to explore the optimal $\nu$, where one task minimizes the attention energy and the other maximizes the distribution probability. Inspired by Nash-MTL (Navon et al., 2022), we model the gradient combination of these two tasks as a bargaining game, solving for the Nash Bargaining Solution. Let

$\{g_j \in \mathbb{R}^d | 1 \leq j \leq K\}$ represent the gradients of $K$ tasks, the optimal gradient combination coefficients $\alpha \in \mathbb{R}_+^K$ satisfy $G^T G \alpha = \frac{1}{\alpha}$, where $G$ is the matrix whose columns are the gradients $g_j$. Nash-MTL uses optimization to approximate the solution for $\alpha$, and we find that when $K = 2$, this equation has a simple analytical solution:

**Corollary 4.2.** *Given* $G = (g_1, g_2) \in \mathbb{R}^{d \times 2}$ *and* $\alpha = (\alpha_1, \alpha_2)^T \in \mathbb{R}_+^2$, *if* $G^T G \alpha = \frac{1}{\alpha}$, *then we have* $\frac{\alpha_1}{\alpha_2} = \frac{||g_2||}{||g_1||}$.

*Proof.* According to $G^T G \alpha = \frac{1}{\alpha}$, we have:

$$\alpha_1^2 ||g_1||^2 + \alpha_1 \alpha_2 g_1^T g_2 = 1, \qquad (17)$$

$$\alpha_1 \alpha_2 g_2^T g_1 + \alpha_2^2 ||g_2||^2 = 1. \qquad (18)$$

By subtracting Equation (18) from Equation (17):

$$\alpha_1^2 ||g_1||^2 = \alpha_2^2 ||g_2||^2, \qquad (19)$$

we can derive the conclusion as $\frac{\alpha_1}{\alpha_2} = \frac{||g_2||}{||g_1||}$. $\qquad \square$

Based on the above proof, we propose apative update rule by formalize $\nu$ as an adaptive parameter for each iteration:

$$\nu = \frac{||\nabla_{z_t} \log(p(z_t))||}{||\nabla_{z_t} \mathcal{E}_{naef}(z_t, m))||}. \qquad (20)$$

This design enables us to effectively mitigate the trade-off at negligible cost, making it more suitable for practical applications.

# 5. Experiments

In this section, we first provide the experimental setup and then conduct both qualitative and quantitative experiments to compare our method with previous SOTA Layout-to-Image methods. Additionally, we perform an ablation study to demonstrate the effectiveness of the proposed approaches.

## 5.1. Experimental setup

*Evaluation Benchmarks.* Akin to prior work (Chen et al., 2024d), we quantitatively evaluate our WinWinLay on COCO2014 (Lin et al., 2014) and Flickr30K (Plummer et al., 2015). For performance evaluation, we leverage YOLOv7 (Wang et al., 2023) for object detection, employing metrics such as AP (Li et al., 2021) to assess the effectiveness of our method in accurately locating and generating objects. Furthermore, the CLIP-s (Radford et al., 2021) is utilized to quantitatively evaluate image-text compatibility, thereby measuring the semantic accuracy of the synthesized images. Additionally, we also use the advantage metric FID (Kynkäänniemi et al., 2023), PickScore (Kirstain et al., 2023) and ImageReward (Xu et al., 2023) to evaluate image quality. Here, we set text template as "A photo of [prompt]" to acquire more realistic results.

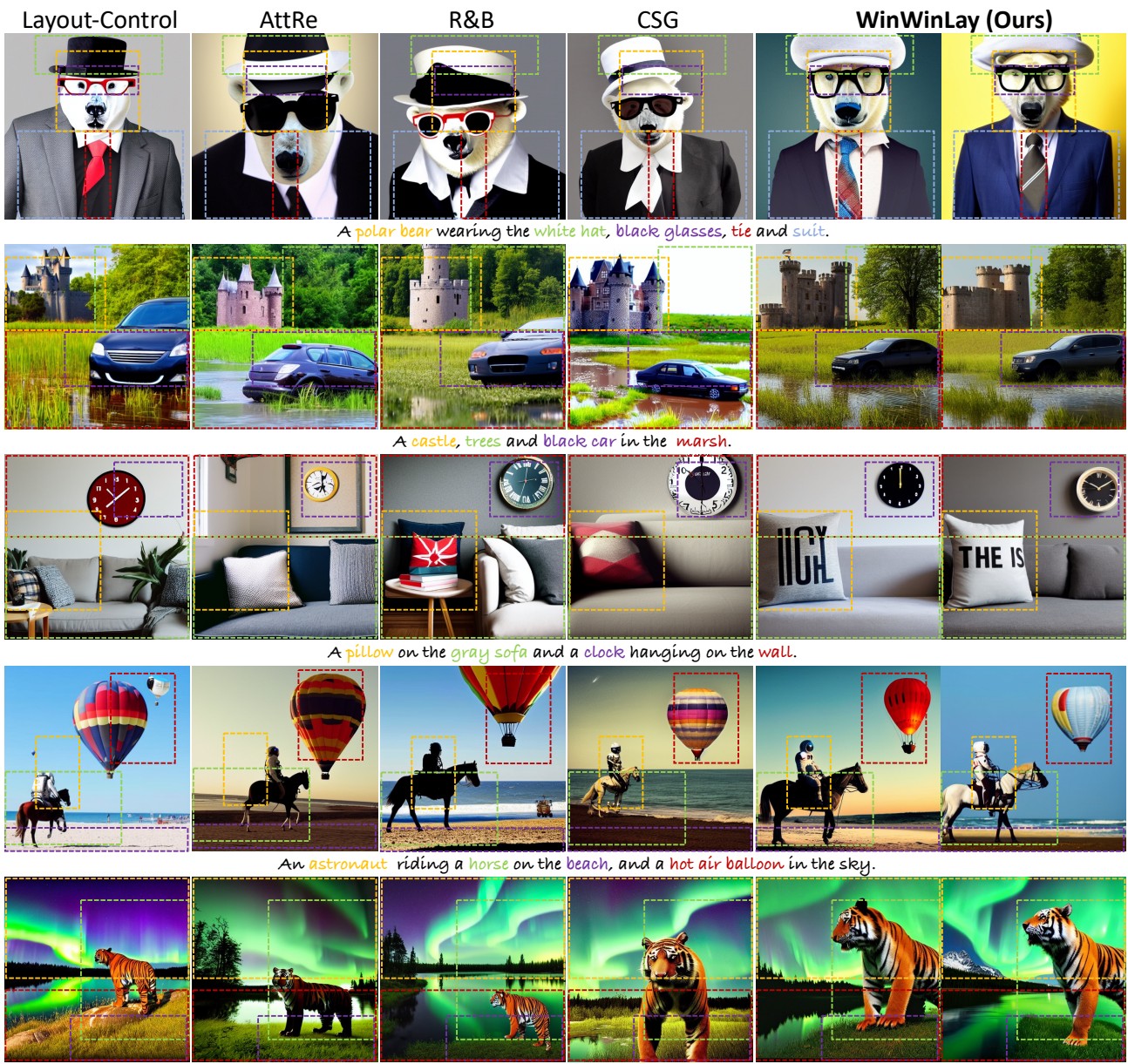

Figure 4: Qualitative comparison of our WinWinLay and state-of-the-art methods. Zoom in for more details.

| Model | COCO2014 | | | | | Flicker30K | | | | | User Study | |
|---|---|---|---|---|---|---|---|---|---|---|---|---|
| | AP↑ | CLIP-s↑ | FID↓ | PickScore↑ | ImageReward↑ | AP↑ | CLIP-s↑ | FID↓ | PickScore↑ | ImageReward↑ | Controllability↑ | Quality↑ |
| Layout-Control (Chen et al., 2024d) | 8.42 | 0.310 | 29.79 | 21.09 | 0.7038 | 14.19 | 0.288 | 29.83 | 20.39 | 0.7016 | 12.1 | 7.5 |
| AttRe (Phung et al., 2024) | 15.51 | 0.296 | 27.51 | 21.23 | 0.7109 | 15.26 | 0.277 | 27.72 | 20.64 | 0.7095 | 18.7 | 22.3 |
| R&B (Xiao et al., 2024) | 17.63 | 0.306 | 28.22 | 21.16 | 0.7071 | 14.80 | 0.291 | 28.18 | 20.58 | 0.7114 | 20.6 | 19.4 |
| CSG (Liu et al., 2024a) | 17.58 | 0.299 | 27.64 | 21.22 | 0.7027 | 15.11 | 0.282 | 27.90 | 20.51 | 0.7049 | 20.9 | 21.0 |
| **Ours** | **19.74** | **0.327** | **26.85** | **21.41** | **0.7218** | **17.28** | **0.309** | **27.04** | **20.85** | **0.7202** | **27.7** | **29.8** |

Table 1: Quantitative comparison of our WinWinLay and state-of-the-art methods.

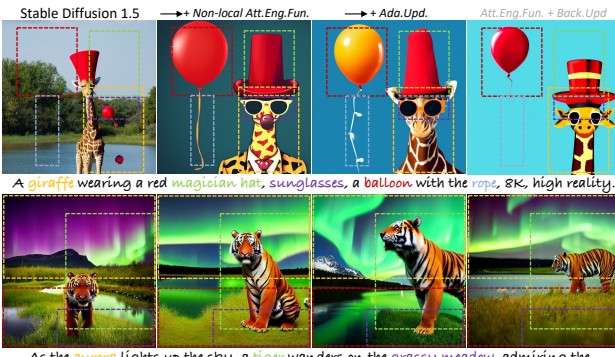

Figure 5: Qualitative ablation on proposed Non-local Attention Energy Function and Adaptive Update. Zoom in for more details.

| Model | COCO2014 | | |
|---|---|---|---|
| | $AP_{50}\uparrow$ | $AP\uparrow$ | CLIP-s$\uparrow$ |
| Att.Eng.Fun. + Back.Upd. | 25.8 | 8.4 | 0.310 |
| **Non-local Att.Eng.Fun.** + Back.Upd. | 44.1 | 17.4 | 0.318 |
| Att.Eng.Fun. + **Ada.Upd.** | 36.7 | 14.9 | 0.324 |
| **Non-local Att.Eng.Fun.** + **Ada.Upd.** | **49.2** | **19.7** | **0.327** |

Table 2: Quantitative ablation on proposed Non-local Attention Energy Function and Adaptive Update on COCO2014.

***Implementation Details.*** We adopt the Stable Diffusion 1.5 (Rombach et al., 2022), pre-trained on the LAION-5B (Schuhmann et al., 2022a), as our base Text-to-Image model. During generation, we employ the DDIM sampler with 50 steps and set the scale guidance to 7.5 for generation. Since layout construction typically occurs during the early stages of denoising, we apply the layout constraint only within the initial 10 steps. The hyperparameters $\rho$ of non-local attention prior is set to 5/0 for max/min, respectively. For adaptive update, we set steps $O$ of Langevin dynamics is set as 4 and signal-to-noise ratio $r$ as 0.06. We observe that these parameters generally work well in most cases, proving the generalizability of WinWinLay. We also point that better results may be obtained with a customized setting, *e.g.*, a larger $\rho$ or more iterations for Langevin dynamics.

### 5.2. Comparison With SOTA Methods

We compare our approach with four representative state-of-the-arts of Layout-Control (Chen et al., 2024d), AttRe (Phung et al., 2024), R&B (Xiao et al., 2024) and CSG (Liu et al., 2024a) to show the advantage of Win-WinLay. All methods are implemented by official codes.

***Quantitative Comparison.*** As presented in Table 1, we first quantitatively evaluate generated images for our test dataset. Compared to methods Layout-Control and AttRe, CSG shows a significant improvement in object placement accuracy. However, we observed in our experiments that it is

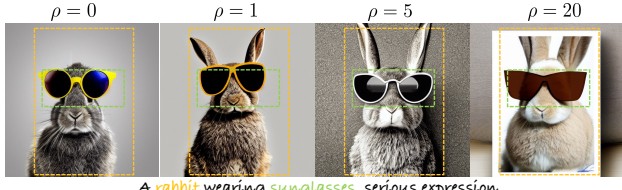

Figure 6: Ablation on hyperparameter of non-local attention prior.

highly sensitive to gradient strength, where higher accuracy often leads to a severe decline in image quality, especially when generating a large number of objects. In contrast, our method consistently outperforms across multiple datasets and evaluation metrics, demonstrating a more robust improvement. Additionally, we resort to user studies to evaluate which method generates results that are most favored by humans. We conducted two user studies on the results in terms of the Controllability and Quality. In the first study, participants are asked to select the images that best align with the given layout. In the second study, participants are tasked with identifying the images that exhibit the most realistic appearance. To ensure clarity and reproducibility, we conducted the user study on Wenjuanxing, a platform similar to Amazon Mechanical Turk. 150 participants evaluated 50 image pairs, yielding 7500 responses per study. Images were shown side-by-side with layout prompts, and both question order and image positions were randomized to avoid bias. As shown in Table 1, over 27.7% of our results are selected as the best in both two metrics, which proves a significant advantage in generation.

***Qualitative Comparison.*** To present a more detailed and visual comparison of our model, we carry out experiments on a smaller hand-crafted dataset with 3-5 objects. For fair comparison, we generate 10 images for each method under the same random seed and subsequently select the optimal image based on the $AP_{50}$ for display. To demonstrate the effectiveness of WinWinLay, 2 images are presented for each case. From the results in Figure 4, we can draw the following conclusions: ($i$) Our method effectively places the target object within the given region while filling the entire bounding box without compromising the natural structure of the object, representing a significant improvement over existing methods. In contrast, other approaches often fail to generate images that faithfully adhere to the layout (*e.g.*, $1^{th}$ row), and parts of the object may collapse into localized regions of the bounding box (*e.g.*, $4^{th}$ row); ($ii$) Our method successfully eliminates the trade-off between control and quality, without compromising the generative capability of the underlying model despite the additional layout constraints. However, existing works focus on layout adherence while neglecting the realism of the generated objects (*e.g.*, $3^{th}$ row). Furthermore, multiple distinct results generated under the same prompt and spatial constraints demonstrate

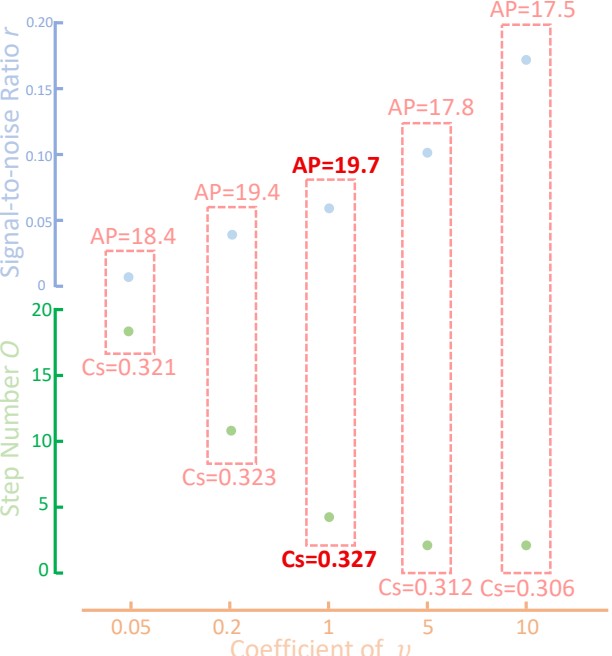

Figure 7: Ablation on hyperparameter of Adaptive Update on COCO2014. Cs denotes CLIP-s metric.

the robustness of WinWinLay, thereby further advancing the progress of Layout-to-Image in practical applications.

### 5.3. Ablation Study

***Effect of Proposed Strategies.*** To further demonstrate the efficacy of the proposed method, we incrementally introduce Non-local Attention Energy Function and Adaptive Update to the baseline model and observe the resulting changes. As shown in Figure 5, Non-local Attention Energy Function significantly enhances the control over the layout, while also ensuring an accurate representation of all target objects. On the other hand, Adaptive Update not only strengthens the spatial placement accuracy but also improves the overall image quality (*e.g.*, "giraffe" appears more realistic). Furthermore, the quantitative results provided in Table 2 align with the visual observations, with Non-local Attention Energy Function achieving a substantial increase in AP and $AP_{50}$, while Adaptive Update further refines the spatial positioning and enhances image quality.

***Hyperparameter of Non-local Attention Prior.*** Previous Attention Energy Function often suffers from the problem of attention collapse to local regions. Hence, non-local attention prior is introduced to constrain the attention to focus on the center of the bounding box and gradually expand to cover the entire box. Here, $\rho$ serves as the hyperparameter that controls the strength of the non-local attention prior. As shown in Figure 6, with the gradual increase of $\rho$, the objects in the image progressively align with the edges of the bounding box, allowing for more precise layout control.

However, when $\rho$ becomes too large, it may lead to unnatural object placements within the bounding box, such as a "rabbit" appearing on a square canvas. In our experiments, we find that setting $\rho$ to 5 generally yields optimal results, which is used across all experiments.

***Hyperparameter of of Adaptive Update.*** Adaptive parameter $\nu$ plays a critical role in the effectiveness of the proposed Adapative Update and impact the overall performance of WinWinLay. In Section 4.2, we analyzed the impact of different $\nu$ on efficiency and performance, proposing an adaptive strategy to significantly reduce the complexity of hyperparameter tuning. To validate its effectiveness, we introduce coefficients of varying magnitudes to the adaptive parameter and conduct grid searches to determine the signal-to-noise ratio $r$ of optimal step size and the number of update steps $O$ for each coefficient. As shown in Figure 7, larger $\nu$ generally require larger step sizes and fewer update steps, which substantially degrade both accuracy and quality. Conversely, smaller $\nu$ has less pronounced effects on performance but significantly increase generation time.

## 6. Conclusion

The paper introduces WinWinLay, a novel training-free framework for Layout-to-Image generation, which achieves significant improvements in layout precision and visual fidelity. Addressing limitations in existing methods, WinWinLay incorporates two innovative components: Non-local Attention Energy Function, which ensures uniform attention distribution within specified layouts while preserving natural object structures, and Adaptive Update, which leverages Langevin dynamics to effectively balance layout control and image quality. Extensive experiments on standard benchmarks demonstrate that WinWinLay outperforms state-of-the-art approaches in both controllability and photorealism, making it a robust and efficient solution for L2I tasks.

## Impact Statement

This project provides a training-free method for layout-controlled image synthesis, enhancing controllability while preserving generative strength of base models; however, like other generative techniques, it may be misused for disinformation, highlighting the need for future research to address ethical risks associated with layout-guided generation.

## Acknowledgements

This paper is supported by National Natural Science Foundation of China under Grants (U23B2012, 12471308), Beijing Natural Science Foundation (1254050), Fundamental Research Funds for the Central Universities, and National Research Foundation, Singapore, under its Medium Sized Center for Advanced Robotics Technology Innovation.

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
