# OpenReview forum: "Control and Realism: Best of Both Worlds in Layout-to-Image without Training"
_ICML.cc/2025/Conference — ICML 2025 poster_

### Official Review · Reviewer_YGgi · 2025-03-12

**Overall Recommendation:** 3

**Summary:**

This work introduces WinWinLay, a novel training-free optimization strategy for layout-to-image generation using text-to-image diffusion models. The paper tackles two main drawback of previous approaches for layout-to-image generation: (1) that the generated objects are often not precisely places within the given bounding boxes, and (2) that there is clear trade-off between the controllability from the layout and the overall quality of the image. First, the authors provide a theoretical analysis on the backward guidance, and propose a non-local attention energy function for better adherence to the bounding boxes. Then, to mitigate the quality-accuracy trade-off, the authors introduce the adaptive update based on Langevin-dynamics. The experimental results show that WinWinLay sets the state-of-the-art on training-free layout-to-image generation.

**Claims And Evidence:**

1. The problem definition of the paper is reasonable. Training-free layout-to-image generation is a practically important application, and there have been limitations in the layout adherence and the image quality, and a trade-off between the two.

2. The authors provide a thorough analysis on the optimization behavior of the attention energy function and why it becomes difficult for the generated object to cover the whole bounding box region. The proposed solution, non-local attention prior, is a intuitive choice to mitigate this. However, to claim its novelty and clear effectiveness compared to previous approaches, further justications on the design of the prior distribution $\tau_u$ and comparisons with previous works is required. For further details on this, please refer to the below section.

3. The authors identify the critical issue of the trade-off between generation (layout) control and image quality. Applying the Langevin dynamics-based update to alleviate this trade-off seems to be a clear and valid approach.

4. While bringing the idea of Nash-MTL seems novel and acts as a key technical detail of WinWinLay, it is hard to understand its effectiveness clearly. Perhaps an ablation study with and without applying this idea could help undertand better.

**Essential References Not Discussed:**

The paper lacks critical references to the previous works on the topic of "training-free layout-to-image generation", which should have also been considered as baselines for quantitative/qualititative comparison. Notably, Attention-Refocusing [1] and R&B [2] were previous works on the same topic, both published at a conference in 2024. These works also aim to address the problem of limited adherence to the bounding box inputs, and the degrated image quality due to the introduction of the backward guidance.

Therefore, without discussions on these prior works and comparisons between the outputs of WinWinLay and these baselines, it would be difficult to claim the novelty and the effectiveness of the proposed approach.

[1] Grounded Text-to-Image Synthesis with Attention Refocusing, Phung et al., CVPR 2024

[2] R&B: Region and Boundary Aware Zero-shot Grounded Text-to-image Generation, Xiao et al., ICLR 2024

**Experimental Designs Or Analyses:**

1. The paper doesn't present sufficient exploration on the previous state-of-the-art methods, and misses out important baselines. Please see the "Essential References" section for this.

2. Since enhancing the image quality is a key objective of WinWinLay, it is expected to show more concrete comparisons on image quality compared to the baselines. Other than the user study, there are multiple widely used metrics to measure the image quality and diversity. I have discussed this in the above section.

**Methods And Evaluation Criteria:**

1. Introducing the "non-local attention prior" for encouraging the object to cover a larger region within the bounding box seems as a reasonable choice. However, defining the prior distribution to be centered at the "centroid" of the bounding box would need a better justification. It seems to be quite a hard constraint when we want an object to be generated within the box. For some objects, it is more intuitive to think that its center of mass is at the lower region of the box (e.g. a car) rather than the center.  Have the authors encountered any failure cases in which enforcing the center to be at the centroid of the box led to degradation of the objects?

2. While one of the key objectives of "non-local attention prior" is to maximize the coverage of the bounding box, there is no discussion and comparison to R&B [1], which also addresses a similar issue by introducing a boundary-aware loss using the Sobel operator. Since R&B is the state-of-the-art method on training-free layout-to-image generation, it is crucial to discuss which aspect WinWinLay has enabled it to obtain results that R&B had failed to achieve.

3. The paper lacks critical baselines for comparison. Please refer to the "Essential References" section for this.

4. For the user study to have validify, the authors should provide more details on its setup. For instance, how were the participants gathered (e.g. Amazon Mechanical Turk), and how many people participated? What was the question format and how many images were presented at each question?

5. In addition to user study, there are multiple metrics that can be used to measure the image quality. FID, PickScore [2] and ImageReward [3] are widely used metrics for this. User study alone seems insufficient to claim the clear advantage of WinWinLay compared to the baselines.

[1] R&B: Region and Boundary Aware Zero-shot Grounded Text-to-image Generation, Xiao et al., ICLR 2024

[2] Pick-a-Pic: An Open Dataset of User Preferences for Text-to-Image Generation, Kirstain et al., NeurIPS 2023

[3] ImageReward: Learning and Evaluating Human Preferences for Text-to-Image Generation, Xu et al., NeurIPS 2023

**Other Comments Or Suggestions:**

The example in Fig. 2 may not be the best choice for claiming that it is important for each object to covered the whole bounding box. Even if the rabbit does not fully fit the box, it is likely to satisfy the user's intentions. If there is an example in which the situation or the story in the image may change when the object doesn't fit the box, it could be a better example showing the importance of maximum coverage.

**Other Strengths And Weaknesses:**

I have discusses the main points in the above sections.

**Questions For Authors:**

Please refer to the above sections.

**Relation To Broader Scientific Literature:**

In the broader field of image generation, enhancing user control is important. Suggesting accurate layout-based controls while preserving the image quality would extend the applicability of text-to-image model to practical use cases in which the users requires precise region control.

**Theoretical Claims:**

Each theoretical claim in the paper is supported by a proof, which seem to be valid.

---

> ### Author Rebuttal · Authors · 2025-03-30
>
> We would like to sincerely thank Reviewer Reviewer YGgi for the thorough and constructive feedback on our manuscript. We are more than happy that the reviewer finds our problem definition is reasonabl, theoretical analysis illustrative, and superior performance. We would like to address the concerns as below.
>
> ***W1. Non-local attention prior.***
>
> **A1.** We acknowledge that for certain object categories—such as coconut trees *(As discussed in lines 80-84)* and cars —the center of mass may deviate from the geometric center of the bounding box due to their asymmetric or elongated structures, making strict centralization suboptimal. To address this, we adopt a decaying schedule for weight $\rho$ of non-local attention prior *(Lines 225–229)*, which applies stronger centralizing guidance during early denoising and gradually relaxes it as generation progresses. This design allows the model to first stabilize attention and then adapt to the object’s natural shape and pose. *As shown in Fig.4*, our method successfully generates well-aligned objects like cars and astronauts without rigidly adhering to box center. Moreover, our *ablation in Fig.6* shows that maintaining a high prior weight throughout can degrade results, confirming the effectiveness of our decay strategy. In summary, the non-local attention prior functions as a soft, adaptive constraint, enabling both precise spatial control and natural object generation across diverse categories.
>
> ***W2. Ablation of Nash-MTL.***
>
> **A2.** Adaptive update is a key component of WinWinLay, designed to balance layout controllability and visual quality with minimal computational overhead. *As shown in Tab.2*, our ablation study demonstrates that incorporating this strategy consistently improves both spatial fidelity and semantic alignment. Moreover, *Fig.7* shows that it reduces the number of update steps by dynamically adjusting optimization directions, leading to faster convergence. We further report actual inference times *in Tab.3*, confirming that adaptive update significantly improves generation efficiency.
>
> ***W3. Example in Fig.2.***
>
> **A3.** Adequate object coverage within the bounding box is crucial for precise and reliable user control. In our evaluation, metrics like AP depend on the IoU between generated objects and target boxes—if an object covers only part of the box, IoU drops, leading to lower AP. This not only affects quantitative results but also indicates a mismatch with user-defined spatial intent, particularly in scenes with dense object layouts requiring fine-grained control.
>
> ***W4. Compared with R&B.***
>
> **A4.**  To clarify the distinctions and advantages of WinWinLay, we provide a focused analysis of the two core components of R&B: Region-aware Loss (RAL) and Boundary-aware Loss (BAL).
>
> (1) RAL leverages the IoU between the predicted box $\hat{\mathcal{B}}$ and the ground-truth box $\mathcal{B}$ to guide object placement. However, this mechanism introduces two critical limitations:
>
> - The non-differentiability of bounding box operations makes gradient-based updates less reliable and harder to tune.
> - Similar to traditional attention energy functions, RAL does not mitigate local bias, leading to incomplete or inaccurate coverage within the target region.
>
> (2) BAL employs a Sobel filter to enhance attention contrast at object boundaries within the box, with the goal of improving object realism. However, it too adopts an energy function-based formulation, and as such, suffers from similar drawbacks:
>
> - Due to the non-differentiability of IoU, BAL reduces to a heuristic energy-based form, which continues to suffer from local bias and incomplete box filling.
> - BAL does not account for the underlying latent distribution $z_{t}$, and therefore cannot effectively navigate the trade-off between layout adherence and image fidelity.
>
> Together, these differences underscore the principled, efficient, and robust nature of WinWinLay in addressing the limitations that R&B is inherently constrained by. We have now included this discussion in revised manuscript.
>
> ***W5. More baselines and metrics.***
>
> **A5.** Please see A1 for Reviewer hxyx.
>
> ***W6. Details of user study.***
>
> **A6.** To ensure clarity and reproducibility, we conducted the user study on Wenjuanxing, a platform similar to Amazon Mechanical Turk. 150 participants evaluated 50 image pairs, yielding 7500 responses per study. For each pair, users answered two questions:
>
> - Which image better matches the bounding box layout?
> - Which image has higher visual quality?
>
> Images were shown side-by-side with layout prompts, and both question order and image positions were randomized to avoid bias. These details have been included in revised manuscript for completeness.
>
> Thanks again for the insightful review. We are happy to discuss any aspects of the manuscript that may require further clarification.

---

> > ### Comment · Reviewer_YGgi · 2025-04-05
> >
> > I appreciate the authors for addressing the raised concerns, especially providing quantitative comparisons with R&B and clarifying the technical contributions of WinWinLay.
> >
> > I have updated my recommendation to "Weak Accept".

---

> > > ### Author Response · Authors · 2025-04-05
> > >
> > > We are truly grateful for your thoughtful and constructive feedback, which has been instrumental in improving our work. Thank you again for your time and valuable input throughout the review process :)

---

### Official Review · Reviewer_QbsF · 2025-03-13

**Overall Recommendation:** 3

**Summary:**

This work aims to achieve high-quality generation for the Layout-to-Image generation task without requiring any training data. It begins by providing a theoretical analysis of existing backward guidance methods and introduces a novel Non-local Attention Energy Function, which enables the model to better respect spatial constraints while maintaining the natural structure of objects. Furthermore, the authors identify that the standard backpropagation update rule can lead to deviations from the pre-trained domain, resulting in visual artifacts. To address this issue, the work proposes a Langevin dynamics-based Adaptive Update scheme, effectively balancing layout adherence and visual realism. Extensive experiments demonstrate the method’s superior performance in both controllability and realism.

**Claims And Evidence:**

Yes

**Essential References Not Discussed:**

No

**Experimental Designs Or Analyses:**

Yes

**Methods And Evaluation Criteria:**

Yes

**Other Comments Or Suggestions:**

No.

**Other Strengths And Weaknesses:**

Strength:
- The paper is clearly structured and easy to follow, providing a smooth and coherent flow of ideas.
- The analysis presented in this work is both comprehensive and insightful. It provides a deep investigation into the limitations of existing methods and introduces effective solutions that significantly improve generation quality and controllability, including Non-local Attention Energy Function and Langevin dynamics-based Adaptive Update scheme. Together, these innovations lead to more controllable and higher-quality generation outcomes.
- Experimental results demonstrate that the proposed method achieves superior performance compared to state-of-the-art techniques, both qualitatively and quantitatively, highlighting its effectiveness.

Weakness:
- The proposed method is evaluated using Stable Diffusion 1.5, which is relatively outdated. Have the authors considered implementing their approach on more recent models, such as Stable Diffusion 3/3.5 or FLUX?

**Questions For Authors:**

The examples shown in the visualizations involve relatively large bounding boxes, which may provide more contextual cues to the model. However, it remains unclear how the method performs with smaller bounding boxes corresponding to small objects (e.g., a mug on a table). Have the authors evaluated the model’s generation quality and spatial controllability under such scenarios?

**Relation To Broader Scientific Literature:**

Layout-to-Image generation has emerged as a prominent research area. This work offers a theoretical analysis of existing backward guidance methods and proposes novel algorithms to address their limitations.

**Theoretical Claims:**

Yes

---

> ### Author Rebuttal · Authors · 2025-03-30
>
> We would like to sincerely thank Reviewer QbsF for the thorough and constructive feedback on our manuscript. We are more than happy that the reviewer finds our paper is clearly structured, comprehensive and insightful analysis, and superior performance. We would like to address the concerns as below.
>
> ***W1. Other base model.***
>
> **A1.**  In line with prior works, we adopt SD1.5 as the default base model in our experiments to ensure a fair and direct comparison with existing training-free Layout-to-Image methods. This choice is consistent with the evaluation protocols followed by representative baselines such as BoxDiff, Layout-Control, and CSG. To further demonstrate the generality and scalability of our approach, we also extend WinWinLay to more recent and powerful SDXL model. As shown in the teaser figure and in additional visualizations provided in the Appendix, our method consistently delivers superior performance under the SDXL setting as well—achieving strong controllability and realism.
>
> ***W2. Small objects.***
>
> **A2.** In fact, both the COCO2014 and Flickr30K datasets used in our experiments naturally contain a significant number of small object instances, such as mouse, coffee cups, and similar items. Therefore, the quantitative results already reported in Tab.1 and Tab.2 implicitly reflect the model’s effectiveness in handling small bounding boxes. To more directly address the reviewer’s concern, we have conducted an additional focused evaluation specifically targeting small-object scenarios. In this experiment, we filtered the datasets to retain only those bounding boxes that occupy no more than 1/8 of the total image area, following a standard definition of small object regions. As expected, all methods exhibit some performance degradation under these more challenging conditions, due to the intrinsic difficulty of generating fine-grained details in small spatial regions. However, our method consistently outperforms all baselines across multiple metrics in this setting as well, demonstrating its robustness, precise spatial control, and superior fine-detail generation capabilities. Specifically, we observe that when the bounding boxes of multiple objects exhibit unnatural relationships—for instance, when a cat and an avocado are assigned bounding boxes of the same size—most existing algorithms suffer from significant performance degradation. In contrast, our method consistently maintains strong performance under such conditions. Visual results are visible on this [link](https://anonymous.4open.science/r/WinWinLay/Small%20objects/Small%20objects.png).
>
> |            Model             | Multidiffusion | BoxDiff | Layout-Control |  CSG  | **Ours**  |
> | :--------------------------: | :------------: | :-----: | :------------: | :---: | :-------: |
> |    COCO2014(AP$\uparrow$)    |      12.2      |   8.1   |      6.9       | 15.7  | **18.2**  |
> |  COCO2014(CLIP-s$\uparrow$)  |     0.267      |  0.281  |     0.279      | 0.274 | **0.296** |
> |   Flicker30K(AP$\uparrow$)   |      10.7      |  13.3   |      13.6      | 14.1  | **16.4**  |
> | Flicker30K(CLIP-s$\uparrow$) |     0.262      |  0.272  |     0.277      | 0.270 | **0.289** |
>
> Thanks again for the insightful review. We are happy to discuss any aspects of the manuscript that may require further clarification.

---

> > ### Comment · Reviewer_QbsF · 2025-04-04
> >
> > Thank the authors for addressing my concerns. I will keep my rating.

---

> > > ### Author Response · Authors · 2025-04-05
> > >
> > > We are truly grateful for your thoughtful and constructive feedback, which has been instrumental in improving our work. Thank you again for your time and valuable input throughout the review process :)

---

### Official Review · Reviewer_hxyx · 2025-03-14

**Overall Recommendation:** 3

**Summary:**

Layout-to-image generation faces two significant challenges: a.) imprecise object localization and b.) the presence of unrealistic artefacts in the final output. To address these issues, the authors propose Win-Winlay, a novel method that incorporates:
1. **Non-Local Attention Energy Function**: This function refines attention scores across regions within each layout box, improving localization accuracy.
2. **Adaptive Update Scheme**: A Langevin dynamics-based adaptive update scheme that generates more realistic outputs.

**Claims And Evidence:**

Yes, the claims are well-supported by thorough evidence in the manuscript.
- Figures 2 and 3 clearly illustrate the shortcomings of existing methods.
- The ablation studies in Section 5.3 provide strong evidence justifying the key design choices.

**Essential References Not Discussed:**

All the major works are cited. Although, following work is not discussed:
1. Phung, Q., Ge, S., & Huang, J. B. (2024). Grounded text-to-image synthesis with attention refocusing. In Proceedings of the IEEE/CVF Conference on Computer Vision and Pattern Recognition (pp. 7932-7942).
        This work uses attention refocusing mechanism and spatial layouts for controlled generation.

**Experimental Designs Or Analyses:**

Yes, the experimental design for ablation studies is sound and is analyzed well in the manuscript.

**Methods And Evaluation Criteria:**

Yes, the proposed method is well-justified and effectively addresses the issues of imprecise localization and unrealistic artefacts in a training-free approach for the Layout-to-Image generation task.

The method is evaluated on the COCO2014 and Flickr30 datasets. YOLOv7 is used for object detection, with AP metrics assessing the accuracy of generated objects. Additionally, CLIP-s evaluates alignment with the conditioning text, and a user study further supports the findings.

**Other Comments Or Suggestions:**

**Typos**:

- L436: "of" is repeated twice in "Hyperparameter of of Adaptive Update".

**Other Strengths And Weaknesses:**

**Strengths**

- **[S1] Paper Writing**: The paper is well-structured, with clear figures that effectively explain the key ideas.
- **[S2] Theoretical Proofs**: The proposed strategies are strongly supported by rigorous theoretical proofs. All the proofs and equations appear to be correct.
- **[S3] Necessity of the Problem Statement**: Guided generation without fine-tuning in diffusion models is an important research area. The proposed method addresses significant limitations in existing work, such as imprecise localization.
- **[S4] Exhaustive Ablation Studies**: The manuscript thoroughly discusses all key design choices. It provides exhaustive analysis on non-local attention and adaptive updates (Figure 5, Table 2), the effect of $\rho$ (Figure 6), and hyperparameters for adaptive updates (Figure 7).


**Weaknesses**

- **[W1] Missing Baselines**: The proposed method does not include comparisons with Attention-Refocusing [R1], InstanceDiffusion [R2], and MIGC [R3]. The authors must compare their method with these approaches for complete evaluation.

- **[W2] Generation of multiple objects**: Can the proposed method generate images for complex text and layouts, such as "Two apples on a plate" or "Three rabbits wearing sunglasses"? If not, the authors should discuss this limitation in the manuscript.

[R1] Phung, Q., Ge, S., & Huang, J. B. (2024). Grounded text-to-image synthesis with attention refocusing. In Proceedings of the IEEE/CVF Conference on Computer Vision and Pattern Recognition (pp. 7932-7942).

[R2] Wang, X., Darrell, T., Rambhatla, S. S., Girdhar, R., & Misra, I. (2024). Instancediffusion: Instance-level control for image generation. In Proceedings of the IEEE/CVF Conference on Computer Vision and Pattern Recognition (pp. 6232-6242).

[R3] Zhou, D., Li, Y., Ma, F., Zhang, X., & Yang, Y. (2024). Migc: Multi-instance generation controller for text-to-image synthesis. In Proceedings of the IEEE/CVF Conference on Computer Vision and Pattern Recognition (pp. 6818-6828).

**Questions For Authors:**

- What is the rationale for not comparing the proposed method with Attention-Refocusing [R1], InstanceDiffusion [R2], and MIGC? (Refer to W1 in Strengths and Weaknesses.)
- Can the proposed method generate multiple objects of the same category? (Refer to W2 in Strengths and Weaknesses.)

If these concerns are addressed, I am willing to increase my score.

**Relation To Broader Scientific Literature:**

The key insight of this paper is that the widely used attention energy function introduces spatial distribution biases, preventing objects from aligning uniformly with layout boxes. To address this, the authors propose a non-local attention prior that redistributes attention scores for better alignment with layout boxes.

Hence, it is helpful for broader scientific literature where the attention scores need to be redistributed based on a spatial region.

**Theoretical Claims:**

Yes, I checked the correctness of the theoretical claims presented in Theorem 4.1, Corollary 4.2 and other Equations in the paper. All the proofs and equations are correct.

---

> ### Author Rebuttal · Authors · 2025-03-30
>
> We would like to sincerely thank Reviewer hxyx for the thorough and constructive feedback on our manuscript. We are more than happy that the reviewer finds our paper is well-structured, theoretical analysis illustrative, and exhaustive ablation studies. We would like to address the concerns as below.
>
> ***W1. More Baselines.***
>
> **A1.**  As our proposed WinWinLay framework operates in a training-free paradigm for Layout-to-Image generation, all methods selected for comparison in our initial experiments,e.g., BoxDiff and CSG, were likewise training-free approaches. In contrast, methods such as [1] and [2] are training-based, requiring supervised learning on paired layout-image data. These methods operate under fundamentally different assumptions and objectives, making a direct comparison less meaningful within the scope of our setting.
>
> Regarding [3], we acknowledge its relevance. Due to strict page limitations, we initially prioritized comparisons with widely recognized baselines and the most recent SOTA method, CSG (ECCV 2024). Here, to address the concern and to offer the more comprehensive evaluation, we have now included additional experiments comparing our method with [3] and [4], both of which follow training-free paradigms more aligned with our setting.
>
> The extended results demonstrate that WinWinLay consistently outperforms these methods across multiple evaluation metrics, highlighting its superior controllability and visual realism. Visual results are visible on this [link](https://anonymous.4open.science/r/WinWinLay/Compared%20with%20more%20SOTAs/Compared%20with%20more%20SOTAs.png). These findings further substantiate the robustness and effectiveness of our proposed method.
>
> | Model(On COCO2014) | AP$\uparrow$ | CLIP-s$\uparrow$ | FID$\downarrow$ | PickScore$\uparrow$[5] | ImageReward$\uparrow$[6] |
> | :------------------: | :----------: | :--------------: | :-------------: | :-----------------: | :-------------------: |
> |    Layout-Control    |    14.19     |      0.288       |      29.83      |        20.39        |        0.7016         |
> |       AttRe[3]       |    15.26     |      0.277       |      27.72      |        20.64        |        0.7095         |
> |        R&B[4]        |    14.80     |      0.291       |      28.18      |        20.58        |        0.7114         |
> |         CSG          |    15.11     |      0.282       |      27.90      |        20.51        |        0.7049         |
> |         Ours         |  **17.28**   |    **0.309**     |    **27.04**    |      **20.85**      |      **0.7202**       |
>
>
> | Model(On Flicker30K) | AP($\uparrow$) | CLIP-s$\uparrow$ | FID$\downarrow$ | PickScore$\uparrow$[5] | ImageReward$\uparrow$[6] |
> | :----------------: | :------------: | :--------------: | :-------------: | :-----------------: | :-------------------: |
> |   Layout-Control   |      8.42      |      0.310       |      29.79      |        21.09        |        0.7038         |
> |      AttRe[3]      |     15.51      |      0.296       |      27.51      |        21.23        |        0.7109         |
> |       R&B[4]       |     17.63      |      0.306       |      28.22      |        21.16        |        0.7071         |
> |        CSG         |     17.58      |      0.299       |      27.64      |        21.22        |        0.7027         |
> |        Ours        |   **19.74**    |    **0.327**     |    **26.85**    |      **21.41**      |      **0.7218**       |
>
> [1] Instancediffusion: Instance-level control for image generation. CVPR2024.
>
> [2] Migc: Multi-instance generation controller for text-to-image synthesis. CVPR2024.
>
> [3] Grounded text-to-image synthesis with attention refocusing. CVPR2024.
>
> [4] R&B: Region and Boundary Aware Zero-shot Grounded Text-to-image Generation. ICLR2024.
>
> [5] Pick-a-Pic: An Open Dataset of User Preferences for Text-to-Image Generation. NeurIPS2023
>
> [6] ImageReward: Learning and Evaluating Human Preferences for Text-to-Image Generation. NeurIPS2023
>
>
>
> ***W2. Multiple instances.***
>
> **A2.** We would like to clarify that our method is capable of handling  textual prompts and layouts involving multiple instances of the same object. This can be achieved by jointly mapping quantifiers and object names (e.g., "two" + "apples") to multiple bounding boxes, and then enforcing center-based generation constraints within each individual bounding box. Visual results are visible on this [link](https://anonymous.4open.science/r/WinWinLay/Multiple%20instances%20generation/Multiple%20instances%20generation.png) and have been added in the revised version.
>
> ***W3. Typos.***
>
> **A3.** We have carefully checked the grammar of the entire manuscript and made corrections.
>
> Thanks again for the insightful review. We are happy to discuss any aspects of the manuscript that may require further clarification.

---

> > ### Comment · Reviewer_hxyx · 2025-04-03
> >
> > Thank you for addressing my questions. All my concerns have been resolved, and I have nothing further to discuss.

---

> > > ### Author Response · Authors · 2025-04-03
> > >
> > > We are truly grateful for your thoughtful and constructive feedback, which has been instrumental in improving our work. We are pleased to hear that your concerns have been fully addressed and would greatly appreciate it if you would consider reflecting this in your final score. Thank you again for your time and valuable input throughout the review process :)

---

### Official Review · Reviewer_z1Kv · 2025-03-17

**Overall Recommendation:** 4

**Summary:**

This paper is a new method in the layout to image domain.

Two key problems this paper tries to address:
* Imprecise location
* Unrealistic artifacts

The core contribution of this paper:
* A non-local attention energy functino
* An adaptive updating mechanism to balance the spatial control and image quality

The paper has a well-organized structure:
- Basic context, introduction, related work
- Problem analysis, theoretical analysis
- Method details
- Evaluation

This structure is easy to follow. And the theoretical analysis part is the most interesting and insightful part.

## Update after rebuttal
The rebuttal addressed my concerns. So I increased the rating.

**Claims And Evidence:**

The claims have been supported by theoretical anaysis, proof and comprehensive evaluations, including quantitative and qualitative experiments.
The overal claim is solid.

**Essential References Not Discussed:**

This paper has discussed relevant papers.

**Experimental Designs Or Analyses:**

- Quantitative
  - Layout accuracy has been evaluated using AP_50/AP
  - Image-text alignment
- Qualitative
  - User study
  - Some qualitative comparing results

**Methods And Evaluation Criteria:**

This method follows existing evaluation critera. These results make sense to illustrate the effectiveness of the propoesd method.

**Other Comments Or Suggestions:**

- The paper figure uses a lot of vspace, affecting the readability.

**Other Strengths And Weaknesses:**

Main advantages:
- The paper writing is well-organized and easy to follow.
- The problems of existing work have been well illustrated in the theoretical analysis sections
- The quantitative evaluation results show promising improvement over existing works.

There are two main weaknesses of this paper:
- The non-local attention prior seems also introduces a new bias: the object seems to be prioritized into the center region.
- The adaptive update rule has already been proposed in several places, e.g. (Taming Transformers for High-Resolution Image Synthesis)

**Questions For Authors:**

The improved non-local attention energy function is designed to solve the spatial distribution biases problem. However, the non-local attention prior seems also introduces a new bias: the object seems to be prioritized into the center region. This may not be a critical problem. But it seems that it's better included in the limitation or discussion sections.

Any other limitations of the proposed method?

**Relation To Broader Scientific Literature:**

This paper introduces some improvements on the layout to image generation problem. Potentially the idea could also helps other training-free methods in T2I domain, e.g. some training-free image editing tasks.

**Theoretical Claims:**

There are two theoretical claims:
- Non-local attention energy function helps overcome the spatial distributino biases
- The adaptive update promotes in-domain updating.

Both of the theoretical claims look correct.

---

> ### Author Rebuttal · Authors · 2025-03-30
>
> We would like to sincerely thank Reviewer z1Kv for the thorough and constructive feedback on our manuscript. We are more than happy that the reviewer finds our writing well-organized, theoretical analysis illustrative, and quantitative improvement promising. We would like to address the concerns as below.
>
> ***W1. Object in the center region.***
>
> **A1.** We would like to clarify that generating objects near the center of the bounding box represents a "positive bias", which is a deliberate design choice intended to enhance generation stability and user controllability. Specifically, placing objects at the center of the bounding box offers finer-grained and more predictable spatial control. In contrast, generating objects near the periphery of the box can introduce several undesirable effects:
>
> - Objects placed near the boundaries are more susceptible to spatial overflow, often leading to leakage beyond the given region;
>
> - Non-central placements inherently introduce positional ambiguity, making it difficult to predict or control which subregion of the box will be emphasized during generation.
>
> Such ambiguity increases uncertainty during optimization and often results in instability or inconsistent outcomes. By encouraging central placement within the bounding box, our method achieves a more robust trade-off between layout adherence and generation reliability. This design also enables users to maintain precise control, especially when dealing with dense or complex scenes. To avoid confusion, we have provided additional clarification in both the Method and Limitation sections of the revised manuscript. We emphasize that this center-focused behavior arises from the prior formulated in Equ.13, where the centrality bias is smoothly modulated by the hyperparameter $\rho$ and decayed over generation. This mechanism is not rigid; it merely guides the model toward stable, interpretable spatial alignment without strictly constraining the center of the object to remain fixed at the center *(As discussed in lines 80-84)*.
>
>
>
> ***W2. Adaptive update rule.***
>
> **A2.** While we acknowledge the use of adaptive updates in prior works such as "Taming Transformers for High-Resolution Image Synthesis", we would like to emphasize that adaptive update rule proposed in our method is conceptually and technically distinct, as elaborated below:
>
> - Theoretical Foundation: The adaptive strategies employed in prior works are largely heuristic, lacking a principled formulation. In contrast, our approach is grounded in multi-task optimization theory. Specifically, we formulate the adaptive update as a Nash Bargaining problem between two competing objectives—layout controllability and visual fidelity—leading to a theoretically justified and analytically solvable solution. This formulation offers a deeper understanding of the trade-off and provides a rigorous basis for gradient balancing.
>
> - Gradient Computation: To mitigate computational burden, prior works typically approximate gradients by restricting updates to the final layer or a limited subset of parameters. Our method, however, naturally yields full gradients for each task without introducing additional overhead, as both objectives are defined over the same latent variables in the denoising process. This makes the adaptive update mechanism not only theoretically sound but also computationally efficient and seamlessly compatible with our training-free setting.
>
>
> Taken together, while the high-level motivation is shared, the underlying methodology, implementation, and theoretical framing of our adaptive update rule differ substantially. We have further clarified this distinction in the revised manuscript.
>
> ***W3. Other Limitaion.***
>
> **A3.** Compared to Text-to-Image methods, our Layout-to-Image approach typically requires more time—for instance, SD1.5 takes 3.17s, while WinWinLay takes 15.01s. Nevertheless, our method remains more efficient than the current SOTA method, CSG, which takes 19.88s. Hence, improving the generation efficiency of our model will be an important focus in future work.
>
> ***W4. Use of Vspace.***
>
> **A4.** We have reorganized the content layout in the revised version to ensure improved visual clarity.
>
> Thanks again for the insightful review. We are happy to discuss any aspects of the manuscript that may require further clarification.

---

> > ### Comment · Reviewer_z1Kv · 2025-04-06
> >
> > Thanks for the rebuttals to address my concerns. I do not have further questions. I lean to accept this paper.

---

> > > ### Author Response · Authors · 2025-04-06
> > >
> > > We are truly grateful for your thoughtful and constructive feedback, which has been instrumental in improving our work. Thank you again for your time and valuable input throughout the review process :)

---

### Decision · Program_Chairs · 2025-05-01

**Decision:**

Accept (poster)

**Comment:**

The paper presents an interesting contribution to the field of training-free layout-to-image generation. The proposed WinWinLay method effectively addresses the limitations of existing approaches from its novel non-local attention energy function and update mechanism. The authors provide theoretical grounding for their method, and demonstrate its effectiveness through comprehensive experimental evaluations. While the reviewers initially expressed some concerns regarding novelty, missing baselines, and evaluation details, they were mostly satisfied with the authors' responses and clarifications provided in the rebuttal.

Based on the reviewers' feedback and the several round discussions, it is a clear case for acceptance. I’d recommend the authors to address the reviewers’ comments in the revision.